# Perspectives of Women with Breast Cancer and Healthcare Providers Participating in an Adherence-Enhancing Program for Adjuvant Endocrine Therapy: A Qualitative Study

**DOI:** 10.3390/curroncol32010045

**Published:** 2025-01-17

**Authors:** Véronique Turcotte, Laurence Guillaumie, Martine Lemay, Anne Dionne, Julie Lemieux, Angéline Labbé, Carolyn Gotay, Line Guénette, Sophie Lauzier

**Affiliations:** 1Population Health and Optimal Health Practices Axis, CHU de Québec-Université Laval Research Center, Hôpital du Saint-Sacrement, 1050 Chemin Ste-Foy, Québec, QC G1S 4L8, Canadalaurence.guillaumie@fsi.ulaval.ca (L.G.); line.guenette@pha.ulaval.ca (L.G.); 2Faculty of Nursing, Pavillon Ferdinand-Vandry, Université Laval, 1050 Avenue de la Médecine, Québec, QC G1V 0A6, Canada; 3Centre des Maladies du Sein, CHU de Québec-Université Laval, Hôpital du Saint-Sacrement, 1050 Chemin Ste-Foy, Québec, QC G1S 4L8, Canadaanne.dionne@pha.ulaval.ca (A.D.); julie.lemieux@crchudequebec.ulaval.ca (J.L.); 4Faculty of Pharmacy, Pavillon Ferdinand-Vandry, Université Laval, 1050 Avenue de la Médecine, Québec, QC G1V 0A6, Canada; 5Oncology Axis, CHU de Québec-Université Laval Research Center, Hôpital du Saint-Sacrement, 1050 Chemin Ste-Foy, Québec, QC G1S 4L8, Canada; 6Université Laval Cancer Research Center, 9, McMahon Street, Québec, QC G1R 3S3, Canada; 7Centre Interdisciplinaire De Recherche en Réadaptation Et Intégration Sociale (CIRRIS), 525 Boulevard Wilfrid-Hamel, Québec, QC G1M 2S8, Canada; 8School of Population and Public Health, University of British Columbia, 2206 E Mall, Vancouver, BC V6T 1Z3, Canada; carolyn.gotay@ubc.ca; 9Équipe de Recherche Michel-Sarrazin en Oncologie Psychosociale et Soins Palliatifs (ERMOS), 2101 Chemin Saint-Louis, Québec, QC G1T 1P5, Canada

**Keywords:** aromatase inhibitors, breast neoplasms, medication adherence, pilot study, qualitative research, randomized clinical trial, survivorship, tamoxifen

## Abstract

Background: Adjuvant endocrine therapy (AET) is prescribed for 5–10 years to women with hormone-sensitive breast cancer to prevent recurrence. However, a significant proportion of women do not adhere to AET. We developed SOIE, a one-year program designed to enhance the AET experience and adherence. SOIE was pilot-tested in a mixed-methods randomized controlled trial. This report presents the experience of women and healthcare providers (HCPs) with SOIE. Methods: A descriptive qualitative study using semi-structured interviews and thematic analysis was conducted with 20 women and 7 HCPs who participated in the program. Results: Most women and HCPs reported high satisfaction with the program. Women felt it addressed their need for information and strategies to manage side effects. They felt supported and developed a more positive attitude toward AET, which contributed to their intention to pursue AET. They perceived that the program helped them navigate the AET experience and reduced their stress or fear regarding AET. HCPs corroborated these benefits. Conclusions: Findings suggest that SOIE can enhance the experience and motivation to pursue the AET treatment by meeting important needs for information, side-effects management, and psycho-emotional support. Programs like SOIE can have benefits beyond adherence by improving patients’ well-being during this crucial long-term treatment.

## 1. Introduction

Adjuvant endocrine therapy (AET) (tamoxifen or aromatase inhibitors (AIs)) for 5 to 10 years is recommended for women with hormone-sensitive breast cancer (60–75% of cases [1,2]) to reduce the risk of recurrence and mortality [3]. However, AET adherence is sub-optimal for a significant proportion of women; 28–59% of women collect less than 80% of their prescribed doses [4] and 31–47% do not persist with AET for the minimally recommended five years [5]. AET non-adherence has been associated with a higher risk of cancer recurrence and mortality [6,7].

Multiple factors affect adherence to AET [8,9,10,11,12,13,14]. A recent meta-analysis involving 33 studies determined that overall AET adherence interventions had small but statistically significant effects [15]. The authors of this meta-analysis concluded that AET adherence-enhancing interventions must be multifaceted to increase the magnitude of their effect [15]. However, to our knowledge, no intervention [15,16,17] combined all of the dimensions known to support AET adherence [8,10,11,13]. These dimensions are: knowledge acquisition, development of a positive attitude towards AET, skill development for managing side effects (e.g., hot flashes, arthralgia, vaginal symptoms), medication-taking routine, and support from healthcare providers (HCPs) and peers with breast cancer [15].

We designed a multifaceted and interprofessional program entitled “SOIE” (*Soutien*, *Outils*, *Information*, *Entraide*—in English *Support*, *Tools*, *Information*, *Mutual Aid*) that aims to improve the AET experience and adherence. The program targets factors known to support AET adherence and comprises four components delivered over the first year following the first AET prescription: (1) an in-person educational group session delivered by a nurse navigator and a pharmacist (in the first weeks of AET initiation); (2) a paper-based patient guide; (3) two nurse-led telephone consultations inspired by motivational interviewing and guided by evidence-based strategies for coping with AET side effects (at 2 and 11 months) [18]; and (4) two online chat sessions facilitated by a nurse navigator, pharmacists, and a social worker (at 5 and 8 months).

We conducted a one-year pilot randomized controlled trial (RCT) among 106 women (intervention 52; control 54) using mixed methods [19]. A description of the intervention development, its components, and results from the quantitative evaluation were previously reported [20]. These results indicated that the program was feasible and greatly appreciated by the participants. We observed a trend, although not statistically significant, toward a positive impact on women’s intention to adhere to AET (primary outcome). SOIE had statistically significant benefits regarding AET knowledge, preparedness to cope with AET-related difficulties, and self-reported AET daily intake, as well as positive trends for adherence as measured by pharmacy records [20].

To contextualize and complement this quantitative evaluation, we conducted a qualitative study with a sample of women assigned to the intervention group and the HCPs involved in the program. We aimed to collect their experiences with and perceived benefits of the program.

## 2. Material & Methods

### 2.1. The SOIE Program Pilot RCT

The SOIE program and its pilot RCT are described elsewhere [20]. Briefly, we conducted a pilot single-center, parallel-group RCT. In addition to usual care, the intervention group received SOIE, while the control group received usual care only. Women were recruited at their first AET prescription for non-metastatic breast cancer at the Breast Disease Center of the CHU de Québec–Université Laval. Psychosocial factors hypothesized to influence AET adherence according to our conceptual model [20] and targeted by the program (i.e., intention to persist with AET, attitude towards AET, subjective norm, perceived behavioral control, AET knowledge, perceived social support, coping planning, anticipated regret, and fear of recurrence) were measured through questionnaires, before randomization, and after 3 and 12 months. Adherence was measured using questionnaires and pharmacy records. The Research Ethics Board (MP-20-2018-4131) approved this study and participants provided written consent.

### 2.2. Qualitative Study

#### 2.2.1. Design and Participants

We conducted a descriptive qualitative study [21] using semi-structured individual interviews among women assigned to the intervention group. A selection of women was made from the 48 women who completed their one-year follow-up based on a purposeful sampling [22] that considered their responses to the questionnaires. We selected potential participants for this qualitative study to ensure that the proportion of women with certain characteristics (e.g., age, type of AET) was similar to that of the entire sample in the pilot RCT. Additionally, we purposively sampled women with varying levels of satisfaction with the program, as reported in their questionnaires, to capture diverse perspectives. All HCPs who delivered the program activities were invited to participate.

#### 2.2.2. Data Collection

A research professional trained in qualitative research (VT—Community Medicine) performed the interviews over the telephone or through videoconference. Questions focused on perceived facilitators and barriers to program participation, perceived usefulness and impacts of the program, and any suggestions to optimize the program (Appendix A).

#### 2.2.3. Data Analysis

Interviews were audio-recorded and transcribed. We performed a thematic analysis [23] using the NVivo software (V12; QSR International). In collaboration with the principal investigator (SL—Anthropology and Epidemiology), the research professional carried out data segmentation and categorization from the first four interviews and elaborated a preliminary codebook using a mixed approach (inductive and deductive) [24]. The development of this preliminary codebook was based on the interview guide, the conceptual model of the SOIE program and allowed the emergence of codes from the corpus [25]. A validation exercise was performed in which a research assistant (GB or AL—Psychology) independently proceeded to the data categorization from excerpts of the same four interviews using the codebook. In case of disagreements, the team discussed the categorization until consensus. The codebook was then refined with the collaboration of senior researchers (SL and L. Guillaumie—Patient Education). This process was conducted eight additional times until all interviews were analyzed. The research professional (VT) and research assistants (GB, AL) created summaries for each code, including exemplary quotes, as well as comparative tables synthesizing the main codes for women and HCPs. These summaries and the interrelationships between codes were discussed during meetings with the research team (SL, L. Guillaumie) and guided the interpretation of the data.

Based on our previous experiences in qualitative research on medication use among cancer patients [18,26,27], we initially estimated that 20 interviews would be sufficient to achieve data saturation. This rigor criterion indicates that new interviews do not provide substantial additional information on the research question [28,29]. Throughout the data collection process, we assessed saturation based on the interviewer’s (VT) summaries and preliminary data analysis, which were discussed with the principal investigator (SL). During this process, we found that 20 interviews were sufficient to reach data saturation. Quotes were translated from French to English by a professional translator.

## 3. Results

Among the 52 women assigned to the SOIE group, 48 completed the one-year follow-up. Of the 21 women and the 9 HCPs invited to participate in an individual interview, 20 and 7 agreed, respectively. Their characteristics, collected through the pilot RCT questionnaires, are presented in Table 1 and Table 2.

### 3.1. Program Appreciation

#### 3.1.1. Overall Program

Most women reported being satisfied with the program and perceived it answered their needs (Table 3, quotation #1). Participants thought their participation in the different components occurred at the right time in their care trajectory, when they needed information or support. In particular, the educational group session in the first weeks of AET initiation enabled women to know about the potential side effects and strategies to overcome them. Women also shared that the nurse-led telephone consultations and the online chat sessions answered their questions or worries and motivated them to pursue AET despite the difficulties. This finding is in line with the nurses’ viewpoint that their telephone consultations occurred at the right moment when women had questions or needed reassurance or motivation (quotation #2).

Most women found the program comprehensive and appreciated the various means used to provide them with information, strategies, and support. Most women and some HCPs thought that the information shared through the various activities was sufficient and relevant for the experience with AET. Women perceived the information received as clear, easy to understand, and well-synthesized. It was deemed credible as it was delivered by HCPs viewed as competent. HCPs found the paper-based patient guide and the algorithms presenting evidence-based strategies for coping with AET side effects useful for their consultations (quotation #3). Both women and HCPs thought the roles of the different HCPs involved in the program were complementary.

However, many women reported that the information presented during the nurse-led telephone consultations or chat sessions was redundant since most of this information was already accessible in the paper-based patient guide.

#### 3.1.2. Educational Group Session

Most women appreciated the educational group session. They found that the friendly and respectful atmosphere facilitated exchanges between participants and made it easy to ask questions. The use of visual representations eased women’s understanding of the information. The video clips presenting testimonies of women who share their experiences with AET were particularly appreciated. Women reported they could identify with these testimonies or have found them reassuring or encouraging (quotation #4). On the other hand, some participants perceived that the content covered during the information session was too basic and would have appreciated receiving more advanced content.

#### 3.1.3. Nurse-Led Consultations

Many women appreciated the nurse’s consultations because they felt these were personalized and well-adapted to their situation (quotation #5). Nurses reported that the consultation guide, inspired by the motivational interview principles, enabled them to personalize their consultations. Nurses also perceived that their consultations met the women’s needs, particularly those experiencing difficulties. Some women shared that they appreciated the nurses’ proactivity during the consultations. As the nurses initiated the call, it made women feel more comfortable asking questions, as they would not have dared to “disturb” the nurse by calling her. Most women perceived that the nurses were concerned about taking good care of them and had an encouraging attitude. The nurses’ kindness, gentleness, and listening skills fostered a climate of trust, allowing women to feel important and express themselves freely without the fear of being judged. Nurses, for their part, found the women receptive, open, and ready to try out strategies to lessen their side effects (quotation #6).

However, about a third of women would have appreciated receiving more information from the nurse during the consultations or more consultations from the nurse.

#### 3.1.4. Online Chat Sessions

Many women appreciated meeting other patients through the online chat sessions and sharing experiences. Some women perceived the tremendous potential of this activity despite the challenges of exchanges based on text only, which sometimes compromised the fluidity (quotation #7). Some HCPs appreciated the platform’s ease of use and the possibility to consult a history of exchanges from previous sessions to tailor their interventions (quotation #8). Some participants were less comfortable with the use of this technology, and this could cause them to interact less during this activity. Some participants found the atmosphere impersonal or unfriendly (quotation #9) and suggested that a video conferencing platform that allows participants to see each other would have made the exchanges livelier and human.

### 3.2. Perceived Impacts of the SOIE Program for Women

#### 3.2.1. Knowledge, Attitude, Motivation, and Coping Strategies

Most women indicated that their participation in SOIE, particularly in the educational group session, enabled them to become aware or understand the reasons to take AET or the importance of taking AET for the entire duration. According to most women and half of the HCPs, participation in the program led women to be better informed on AET or understand AET better, giving them a sense of trust (quotation #10).

Almost all women and HCPs found the paper-based patient guide very useful. It served as a reference when women had questions or concerns (quotation #11), or it was used to explain their experience with AET to their loved ones. Women and HCPs thought that the information received led women to have a more positive attitude or a better acceptance of AET, and this contributed to their intention to continue to take AET despite side effects (quotation #12).

Many women and HCPs reported that the information and tools provided in the SOIE program helped reduce stress, fear, or anxiety regarding AET or cancer recurrence, secured the participants regarding the risks associated with taking AET, or brought greater confidence in AET. Almost all women and HCPs thought it led women to realize there are solutions to AET side effects and to feel better prepared (quotation #13). Moreover, many women and HCPs thought the SOIE program helped support women in making informed decisions regarding AET.

Some women perceived that their participation in the program did not have any impact, as they were already well-informed and motivated to take AET or intended to take AET for the entire duration.

#### 3.2.2. Support from Healthcare Providers and Peers

Most women perceived the support received during the program helped them get through the AET experience. Almost all women and HCPs thought the program enabled participants to access HCPs in case of difficulties or worries with AET and benefit from follow-ups, support, and encouragement from HCPs. Women indicated that accessing HCPs was reassuring or helped them continue with AET (quotation #14).

According to almost all women and HCPs, participating in the SOIE program, particularly in the chat sessions, enabled women to help each other, to share their experiences or tips to deal with AET side effects, or to encourage each other. Most women and many HCPs thought that the nurse-led consultations and online chat sessions, made participants feel that they were not alone, isolated, or abandoned. Some women also felt valued by helping other women (quotation #15).

Most women reported that their participation in the SOIE program enabled them to normalize their experience by knowing that other women had a similar experience with AET. The feeling of not being alone to experience side effects and feeling “normal” was a source of motivation (quotation #16).

However, some women reported that participating in the educational group or online chat sessions led to questioning AET-taking or feeling insecure, stressed out, or worried about AET, due to the negative attitude of some participants, the focus on difficulties encountered with AET or the sharing of anxiety (quotation #17).

### 3.3. Perceived Impacts of the SOIE Program for HCPs

According to some HCPs, SOIE could reduce their workload regarding the transmission of information on AET. It could also reduce the number of calls about AET at the Center because women would have had an opportunity to ask questions and receive tools to manage side effects by themselves (quotation #18). This viewpoint was also shared by some women (quotation #19).

Some women and one HCP perceived that the SOIE program would be an excellent addition to the follow-up carried out by the HCPs at the Center and would fit well in the continuum of care (quotation #20). However, most HCPs perceived that there would be difficulties integrating the program activities in the same format into their current practice due to staff shortages, work overload, and the large number of patients followed at the Center.

Some HCPs perceived that their involvement in the program enabled them to improve or update their consultations with breast cancer patients who have AET. Others felt it helped them better understand what patients go through when taking AET. Some HCPs reported that providing consultations using motivational interviewing principles or facilitating chat sessions enabled them to experience new ways to intervene so they can transfer into their current practice with patients with breast cancer and in other care contexts. Some HCPs also perceived that collaborating with other types of HCPs was an enriching experience.

## 4. Discussion

This qualitative evaluation of the SOIE program sheds new light on the potential of adherence-enhancing multifaceted interventions since very few studies have evaluated, from the perspective of women, the specific mechanisms by which interventions can improve their experience and adherence with AET [30,31,32].

The qualitative findings obtained are in line with the quantitative results issued from our pilot RCT [20] but give us a better understanding of how interventions such as SOIE may improve knowledge and skills, feelings of being supported, and medication-taking behavior. These three dimensions are now discussed.

A considerable number of women expressed how much they have appreciated and benefitted from the SOIE educational components, including information on AET, its side effects, and potential strategies for dealing with difficulties. Benefits regarding knowledge and coping planning with side effects were also observed in the quantitative evaluation of SOIE [20]. Regarding the modalities of providing such educational components, participants’ comments indicate that it should be provided by a credible source, presented in different formats (e.g., verbal, written), and through several modalities (e.g., individual consultation and group session). This qualitative study also enables us to better understand which educational content has the potential to benefit patients. Participants reported that understanding why AET is important and being informed in advance of side effects and coping strategies enabled them to make informed decisions, stay motivated, feel reassured, and be prepared for potential difficulties. These factors were also identified as influencing AET adherence in observational studies [8,9,10,13]. However, based on participants’ comments, it is also important to avoid information redundancy as well as to strike the right balance between providing too basic and too detailed information.

Women’s appraisal that feeling supported was an essential part of their experience with AET is in line with previous work [8,10,18,33]. Participants mentioned the different types of support they received from the program: at the information-seeking level, notably through educational activities; at the practical level, through personalized advice when they encountered side effects; and at the emotional level, when they needed to be listened to without judgment and reassured. In particular, some participants mentioned that they felt supported by the nurse navigators during telephone consultations and appreciated the pro-activity of these calls, the personalized and non-judgmental approach, and the caring atmosphere during these consultations. Some of these principles are at the heart of the motivational interviewing approach that inspired the development of this program component [18,34]. The feeling of being supported also stemmed from peer support, a source of support that is less well documented in the literature on AET adherence [15]. Women mentioned that being in contact with other women who had gone through the same experience, whether through the video testimonials, the group education session, or the online exchange sessions, made them feel less isolated, normalized their own experience, reassured them, and kept them motivated toward AET. Some women mentioned that it had been rewarding for them to be able to help other women, a positive benefit that we had not anticipated. Based on these comments and the recommendation of a recent systematic review [15], the integration of peer support in AET adherence-enhancing interventions seems to be an avenue worth exploring. Our findings also point to potential areas of program improvements concerning group activities, such as using videoconferencing platforms (which have increased in popularity since the COVID-19 pandemic [35]) and preparing a more detailed plan to manage the sharing of more negative experiences [36].

It was surprising to find that participants’ comments about the program focused mainly on their well-being, peace of mind, and experience with AET, rather than on issues associated with AET adherence *per se*. Even considering that women in the first year of treatment may be particularly motivated to take their AET as prescribed— median proportion of days covered by an AET was 96% in our pilot RCT [20], we can draw at least two important lessons from our qualitative findings. First, some comments indicate that the program has enabled certain women to develop a more positive attitude towards AET, make informed decisions, and remain motivated should they encounter difficulties. These program outcomes would hopefully support AET adherence in the long term. Second, although the main aim of the SOIE program was to improve treatment adherence, women mentioned repeatedly and in different ways that the program had met significant psycho-emotional needs, such as feeling reassured about AET and the risk of cancer recurrence, feeling less alone at this stage of the care trajectory, and feeling prepared. Although adherence to AET is of importance in terms of reducing the risk of breast cancer recurrence [7], programs that enable patients to live peacefully through this critical stage of the care trajectory, for which few structured services are currently available, seem to respond to unmet needs and represent an important goal in itself.

HCPs who participated in the program reported most of the benefits identified by the women as well as specific benefits stemming from their participation. Examples include the benefits for their own practice through continuing education on AET, application of motivational interviewing principles, and access to AET resource materials. They also anticipated benefits such as fewer calls to the Center if women were better informed about AET through programs like SOIE. While the program was viewed positively by participating HCPs, the implementation challenges they identified should be addressed. The results of our RCT, both quantitative and qualitative, suggest that the SOIE program’s activities are feasible in research settings with a limited number of participants. However, offering the program to all women receiving AET in a moderate-to-high-volume center could be challenging. Targeting women likely to face greater difficulties and using technology, such as providing some program components through a website, could help reduce demands on resources.

### Limitations

This qualitative study has its limitations. The women were selected based on some characteristics and their answers to the questionnaires. We purposely invited the women who reported having some dissatisfaction with specific aspects of the program to gain a diversity of experiences. Therefore, the proportion of women unsatisfied is higher in this qualitative evaluation than in the entire sample of patients included in the pilot RCT. In addition, the interviews took place at the end of the one-year follow-up to avoid influencing the answers to the questionnaires used for the quantitative evaluation. However, participants could share their global experience with the program and a vast majority discussed specific program components.

## 5. Conclusions

This qualitative evaluation of the SOIE program has enabled us to delve deeper into the mechanisms by which this type of program can provide benefits in terms of well-being with AET and could potentially improve adherence. Our findings indicate that interventions to enhance the experience with AET and adherence to this treatment must be multifaceted, covering both information, strategies to mitigate difficulties, and the emotional and well-being aspects that seem to be at the heart of the experience of the women we met. However, implementing such interventions needs to be adapted to the context of current clinical practice, where resources are sometimes limited. Future studies could investigate the long-term impact of programs such as SOIE offered early in the treatment trajectory, as well as the role played by the psycho–emotional factors in AET adherence. Finally, our qualitative evaluation enabled us to identify important dimensions of the AET experience that would have been difficult to capture in a purely quantitative evaluation. This underlines the need for developing new scales to document the impact of our programs on the psychological and emotional dimensions of the AET experience, which are at the heart of patients’ concerns.

## Figures and Tables

**Table 1 curroncol-32-00045-t001:** Women’s sociodemographic characteristics (n = 20).

	n	%
Age		
40–49	2	10
50–59	8	40
60–69	7	35
70–79	2	10
80–89	1	5
Higher level of education		
Secondary school	4	20
College degree ^1^	4	20
Professional diploma	3	15
Bachelor degree	4	20
Masters’ or PhD degree	5	25
Type of AET ^2^		
Aromatase inhibitors	13	65
Tamoxifen	4	20
Both	3	15
AET adherence		
Took AET during the study	19	95
Had stopped to take AET during the study	1	5

^1^ In the province of Quebec, “college” refers to pre-university or vocational studies. ^2^ AET: adjuvant endocrine therapy.

**Table 2 curroncol-32-00045-t002:** Healthcare providers’ sociodemographic characteristics (n = 7).

		n
Sex		
	Man	0
	Woman	7
Age		
	25–34	1
	35–44	2
	45–54	3
	55–64	1
Profession		
	Nurse navigator	3
	Pharmacist	3
	Social worker	1
Higher level of education		
	Bachelor degree	4
	Master degree	3
Years of practice in their profession		
	Mean	19.6
Years of practice at the breast cancer center		
	Mean	12.8
Years of practice with women with breast cancer		
	Mean	14.4
Estimated proportion of patients with breast cancer in their caseload		
	Mean	84.2

**Table 3 curroncol-32-00045-t003:** Exemplar quotations from interviewed women and health care providers.

Program Appreciation
**Overall program** Quotation #1: “*So it went well. Really. I have nothing but good things to say about SOIE. I loved it. If I had to start all over again tomorrow and somebody were to ask, “Uh, should I do it or not?” I’d tell them, “Go for it. Do it. It’s a huge help. It gives you the support you need, when you need it, because this is tough [AET].*” (Woman 113) Quotation #2: “*And the last one [telephone consultation], the second one, you know, um, it happens around the one-year mark. After a year, we notice that demotivation sets in with our patients. So, this program allows us to check on them and see if they’re ready for the next steps.*” (Healthcare Provider 01) Quotation #3: “*[…] I was better prepared and it showed, even in the advice I gave, because I found the document to be, comprehensive and up to date. It was a huge help.*” (Healthcare provider 05)
**Educational group session** Quotation #4: “*We saw that they [patients sharing testimonies] were faring. There were difficult moments and some easier ones, but there was a light at the end of the tunnel.*” (Woman 089)
**Nurse-led consultations** Quotation #5: “*Getting help from a nurse who knows your symptoms and your medication, and who gives you solutions adapted to YOUR situation, YOUR life, YOUR routine, and YOUR personality—that’s HUGE. Having access to this resource was a big plus.*” (Woman 006) Quotation #6: “*The women were very receptive. […] There was a willingness to rally together and pull through. […] If a problem came up, they were open to trying other strategies, trying something to get better.*” (Healthcare provider 09)
**Online chat sessions** Quotation #7: “*Sometimes we didn’t have time to read what the others have written. We were getting ready to write a comment, but it was too late.*” (Woman 094). Quotation #8: “*I found it really easy to access (…) It was easy to use, it was still intuitive. It looks like what we’re used to using. Even the older patients were fine with it. (…) I really liked it as a platform*”. (Healthcare provider 07)” Quotation #9: “*It was just a computer. I asked it questions and got answers, but not much of a human interaction, though.*” (Woman 006)
**Perceived impacts of the SOIE program for women**
**Knowledge, attitude, motivation and coping strategies** Quotation #10: “*I liked (…) that they took the time to really explain the side effects of the antihormone. […] You know what to expect and that’s reassuring. […] It helps build trust.*” (Woman 089) Quotation #11: “*We learned that we could get such and such side effects. It was all in the guide, so we had resources! It helps a lot at the start. You don’t feel like you’re in the dark. You get a better understanding. It’s important to be well informed*”. (Woman 113) Quotation #12: “*I mean cancer is scary, and this program […] helped me stay positive. […] If I hadn’t had all this information, I probably would have given up on the pill after the first side effects.*” (Woman 209) Quotation #13: “*I felt like I had more tools to help me get through this… Between the information, the video, the information booklet, and other stuff, I felt like I came out more prepared, you know?*” (Woman 199)
**Support from healthcare providers and peers** Quotation #14: “*Um, so (sigh) having a person on the other end of the line who understood what we were going through and could guide us through the steps too was reassuring. You know, for me it’s a matter of bringing comfort and information to this whole journey.*” (Woman 004) Quotation #15: “*Being able to share our ways of getting through this with others is really great and so rewarding.*” (Woman 113) Quotation #16: “*Knowing that I’m not alone in this was a great source of motivation. Seeing that there were other people going through this, living with this, just like I was, you know?*” (Woman 043) Quotation #17: “*In times like these, we didn’t need any more negative energy in our lives. We wanted good vibes because there was a LOT of negativity.*” (Woman 138)
**Perceived impacts of the SOIE program for HCPs**
Quotation #18: “*You know, if one day there was a general training out there on hormone therapy, I think it […] would lead to fewer calls. […] I think that if something like that […] eventually became like a “prerequisite,” patients would […] be better informed and have all the tools they needed to manage these side effects. And that would also help professionals.*” (Healthcare provider 01) Quotation #19: “*I think that if I didn’t have all the information, I would have been making calls left and right to find out what was going on because I wouldn’t have understood what was happening to me.*” (Woman 113) Quotation #20: “*The program ensured the continuity of the process since we were very well supported there. […] You know, from the time you’re diagnosed until your last radiation treatment, you have all the support you need, but then you’re left with nothing. That’s where the program comes in. […] It gives you the push you need to keep going.*” (Woman 199)

## Data Availability

The datasets generated during the current study are available from the corresponding author upon reasonable request.

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
