# Peer review of "Perspectives of Women with Breast Cancer and Healthcare Providers Participating in an Adherence-Enhancing Program for Adjuvant Endocrine Therapy: A Qualitative Study"

_curroncol, 2025, doi:10.3390/curroncol32010045_

Round 1
Reviewer 1 Report
Comments and Suggestions for Authors
The article presents a qualitative study of the effects of an intervention program to improve adherence to an Adjuvant endocrine therapy in women with breast cancer. The intervention proposal seems interesting in terms of the results presented, however it is not described in the methodology nor is there supplementary material with the description of the sessions and their objectives, which would also help to understand the results. Also in the introduction they could better explain the problem as well as similar interventions, if they exist, and how they designed their intervention based on the state of the art. Finally, in the discussion they could take elements from Table 3 to strengthen their conclusions.
Author Response
AUTHORS’ RESPONSE (CURRONCOL-3390342)
Comments and Suggestions for Authors
—Reviewer 1
We thank the Reviewers for their helpful comments and suggestions. In the following, the Reviewers’ comments are reproduced integrally, are numbered, and appear in bold text. Our responses follow each comment. Any text cited directly from the manuscript is in quotation marks, while any changes made to the text submitted initially are in italics. Changes made to the manuscript in response to Reviewers’ comments and suggestions are shown in track changes in a revised version of the manuscript.
The article presents a qualitative study of the effects of an intervention program to improve adherence to an adjuvant endocrine therapy in women with breast cancer.
The intervention proposal seems interesting in terms of the results presented, however it is not described in the methodology nor is there supplementary material with the description of the sessions and their objectives, which would also help to understand the results.
Thank you for your interest in our intervention. The development of the intervention (i.e., the profile of the committee members who co-constructed the program, the use of the Intervention Mapping approach, and the conceptual model used), as well as the components of the program (i.e., content, objectives, timing, providers, and duration), are presented in a previously published article (Assan O. et al., J Cancer Surviv. 2024 May 4. doi: 10.1007/s11764-024-01599-y). We have now clarified where readers can find information on the development and content of our program in the Introduction: “A description of the intervention development, its components, and results from the quantitative evaluation were previously reported20 (lines 71-72).” We have also retained the following sentence in the Methods: “The SOIE program and its pilot RCT are described elsewhere20” (lines 85).
Reference 20: Assan O, Memoli V, Guillaumie L, Turcotte V, Lemay M, Dionne A, et al. Pilot randomized controlled trial of a program to enhance experience and adherence with adjuvant endocrine therapy among women with non-metastatic breast cancer: 12-month quantitative results. J Cancer Surviv. 2024:Epub ahead of print. doi: 10.1007/s11764-024-01599-y
Also in the Introduction they could better explain the problem as well as similar interventions, if they exist, and how they designed their intervention based on the state of the art.
We made some additions to the Introduction in order to better explain the problem:
We have added information and references to the fact that non-adherence has been associated with increased risks of recurrence and mortality: “AET non-adherence has been associated with a higher risk of cancer recurrence and mortality6,7” (lines 49-50).
Reference 6: Eliassen FM, Blåfjelldal V, Helland T, Hjorth CF, Hølland K, Lode L, et al. Importance of endocrine treatment adherence and persistence in breast cancer survivorship: a systematic review. BMC Cancer. 2023;23(1):625. doi: 10.1186/s12885-023-11122-8
Reference 7: Inotai A, Agh T, Maris R, Erdosi D, Kovacs S, Kalo Z, et al. Systematic review of real-world studies evaluating the impact of medication non-adherence to endocrine therapies on hard clinical endpoints in patients with non-metastatic breast cancer. Cancer Treat Rev. 2021;100:102264. doi: 10.1016/j.ctrv.2021.102264
We have added information and references to the fact that AET adherence is influenced by multiple factors: “Multiple factors affect adherence to AET8-14” (lines 51). The factors influencing AET adherence that could be modified by an intervention are presented in the following sentence: “However, to our knowledge, no intervention combined all of the dimensions known to support AET-adherence. These dimensions are: knowledge acquisition, development of a positive attitude towards AET, skill development for managing side effects (e.g. hot flashes, arthralgia, vaginal symptoms), medication-taking routine, and support from healthcare providers (HCP) and peers with breast cancer” (lines 54-59).
Reference 8: Lambert LK, Balneaves LG, Howard AF, Gotay CC. Patient-reported factors associated with adherence to adjuvant endocrine therapy after breast cancer: an integrative review. Breast Cancer Res Treat. 2018;167(3):615-33. doi: 10.1007/s10549-017-4561-5
Reference 9: Paranjpe R, John G, Trivedi M, Abughosh S. Identifying adherence barriers to oral endocrine therapy among breast cancer survivors. Breast Cancer Res Treat. 2019;174(2):297-305. doi: 10.1007/s10549-018-05073-z
Reference 10: Toivonen KI, Williamson TM, Carlson LE, Walker LM, Campbell TS. Potentially modifiable factors associated with adherence to adjuvant endocrine therapy among breast cancer survivors: A systematic review. Cancers (Basel). 2020;13(1):107. doi: 10.3390/cancers13010107
Reference 11: Moon Z, Moss-Morris R, Hunter MS, Carlisle S, Hughes LD. Barriers and facilitators of adjuvant hormone therapy adherence and persistence in women with breast cancer: a systematic review. Patient preference and adherence. 2017;11:305-22. doi: 10.2147/ppa.s126651
Reference 12: Yussof I, Mohd Tahir NA, Hatah E, Mohamed Shah N. Factors influencing five-year adherence to adjuvant endocrine therapy in breast cancer patients: A systematic review. Breast. 2022;62:22-35. doi: 10.1016/j.breast.2022.01.012
Reference 13: Xu H, Zhang XJ, Wang DQ, Xu L, Wang AP. Factors influencing medication-taking behaviour with adjuvant endocrine therapy in women with breast cancer: A qualitative systematic review. J Adv Nurs. 2020;76(2):445-58. doi: 10.1111/jan.14253
Reference 14: Peddie N, Agnew S, Crawford M, Dixon D, MacPherson I, Fleming L. The impact of medication side effects on adherence and persistence to hormone therapy in breast cancer survivors: A qualitative systematic review and thematic synthesis. Breast. 2021;58:147-59. doi: 10.1016/j.breast.2021.05.005
We have provided more information on previous interventions (i.e. number of interventions and their effect for AET adherence): “A recent meta-analysis involving 33 studies determined that overall AET adherence interventions had small but statistically significant effects15. The authors of this meta-analysis concluded that AET adherence-enhancing interventions must be multifaceted in order to increase the magnitude of their effect” (lines 51-54).
Reference 15: Bright EE, Finkelstein LB, Nealis MS, Genung SR, Wrigley J, Gu HCJ, et al. A systematic review and meta-analysis of interventions to promote adjuvant endocrine therapy adherence among breast cancer survivors. J Clin Oncol. 2023;41(28):4548-61. doi: 10.1200/jco.23.00697
Regarding the development of our intervention, we now refer more explicitly to our previously published paper that describes the program (please see our Response to Comment #1 from Reviewer #1). In this published paper, we explain that the program was co-created by researchers with expertise in medication adherence and breast cancer, clinicians, and women prescribed AET. We also describe how the program was developed using the Intervention Mapping approach, with the Theory of Planned Behavior as the conceptual model. Additionally, we explain that we chose to pilot-test the program in a mixed-methods pilot randomized controlled trial, following the Medical Research Council (MRC) framework for complex interventions.
Finally, in the discussion they could take elements from Table 3 to strengthen their conclusions.
We were unsure whether the Reviewer was requesting the direct use of participants' quotes, as presented in Table 3, in the discussion. It is uncommon to include results verbatim in the discussion; however, we have provided our interpretation of these results. For example, we discussed the three main types of benefits of the program, as reported by the participants and presented in the results section: “The qualitative findings obtained are in line with the quantitative results issued from our pilot RCT20 but give us a better understanding of how interventions such as SOIE may improve knowledge and skills, feeling of being supported, and medication-taking behavior. These three dimensions are now discussed.” (lines 276-279). Elements of the conclusion are also derived from these results: “Our findings indicate that interventions to enhance the experience with AET and adherence to this treatment must be multifaceted, covering both information, strategies to mitigate difficulties, and the emotional and well-being aspects that seem to be at the heart of the experience of the women we met” (lines 358-361).
We realize that Table 3 was inadvertently truncated during the editing process. Unfortunately, quotes 10 to 20 related to the program’s benefits as reported by the participants were not included in the copy provided to the reviewers, which may have caused some confusion. We have corrected this, and Table 3 is now complete in this revised version of the manuscript.
Reviewer 2 Report
Comments and Suggestions for Authors
Your manuscript addresses a critical issue in hormone therapy for breast cancer patients, specifically concerning the necessity of long-term hormonal treatment adherence. It is particularly noteworthy that maintaining medication adherence motivation during post-surgical hormone therapy appears to significantly influence subsequent outcomes. Through your qualitative evaluation of previously conducted comparative trials, you have elucidated narrative oncological practices among patients and healthcare providers. This evaluation presumably provides valuable insights into patient care and survivorship.
I would like to raise the following points:
1. While lines 242-244 indicate that many healthcare providers recognize difficulties in integrating this program into current clinical practice, this challenge is not adequately addressed in the discussion section. Although the conclusion touches upon future directions, this matter warrants more detailed examination within the discussion section.
2. Regarding lines 120-122, the methodology for calculating the initial target sample size requires clarification. Furthermore, a more detailed justification is needed to demonstrate that this sample size is sufficient to validate the research.
3. In section "2.1 The SOIE program pilot RCT," while you have provided a concise summary of previous papers, further condensation might enhance clarity of the description.
Author Response
AUTHORS’ RESPONSE (CURRONCOL-3390342)
Comments and Suggestions for Authors
—Reviewer 2
We thank the Reviewers for their helpful comments and suggestions. In the following, the Reviewers’ comments are reproduced integrally, are numbered, and appear in bold text. Our responses follow each comment. Any text cited directly from the manuscript is in quotation marks, while any changes made to the text submitted initially are in italics. Changes made to the manuscript in response to Reviewers’ comments and suggestions are shown in track changes in a revised version of the manuscript.
Your manuscript addresses a critical issue in hormone therapy for breast cancer patients, specifically concerning the necessity of long-term hormonal treatment adherence. It is particularly noteworthy that maintaining medication adherence motivation during post-surgical hormone therapy appears to significantly influence subsequent outcomes. Through your qualitative evaluation of previously conducted comparative trials, you have elucidated narrative oncological practices among patients and healthcare providers. This evaluation presumably provides valuable insights into patient care and survivorship. I would like to raise the following points:
While lines 242-244 indicate that many healthcare providers recognize difficulties in integrating this program into current clinical practice, this challenge is not adequately addressed in the discussion section. Although the conclusion touches upon future directions, this matter warrants more detailed examination within the discussion section.
We have added a paragraph to discuss the finding from the HCPs more specifically, including implementation challenges: “Regarding the HCPs who participated in the program, they reported most of the benefits identified by the women as well as specific benefits stemming from their participation. Examples include benefits on their own practice through continuing education on AET, application of motivational interviewing principles, and access to AET resource materials. They also anticipated benefits such as fewer calls to the Center if women were better informed about AET through programs like SOIE. While the program was viewed positively by participating HCPs, the implementation challenges they identified should be addressed. The results of our RCT, both quantitative and qualitative, suggest that the SOIE program’s activities are feasible in a research setting with a limited number of participants. However, offering the program to all women receiving AET in a moderate to high volume Center could be challenging. Targeting women likely to face greater difficulties and using technology, such as providing some program components through a website, could help reduce demands on resources” (lines 333-344).
Regarding lines 120-122, the methodology for calculating the initial target sample size requires clarification. Furthermore, a more detailed justification is needed to demonstrate that this sample size is sufficient to validate the research.
We have added details on the initial estimation of our sample size: “Based on our previous experience in qualitative research on medication use among cancer patients26-29, we initially estimated that 20 interviews would be sufficient to achieve data saturation.” (lines 136-138).
Reference 26: Gagne M, Lauzier S, Lemay M, Loiselle CG, Provencher L, Simard C, et al. Women with breast cancer's perceptions of nurse-led telephone-based motivational interviewing consultations to enhance adherence to adjuvant endocrine therapy: a qualitative study. Support Care Cancer. 2022;30(6):4759-68. doi: 10.1007/s00520-021-06692-x
Reference 27: Camiré-Bernier É, Nidelet E, Baghdadli A, Demers G, Boulanger MC, Brisson MC, et al. Parents' Experiences with Home-Based Oral Chemotherapy Prescribed to a Child Diagnosed with Acute Lymphoblastic Leukemia: A Qualitative Study. Curr Oncol. 2021;28(6):4377-91. doi: 10.3390/curroncol28060372
Reference 28: Guillaumie L, Ndayizigiye A, Beaucage C, Moisan J, Grégoire JP, Villeneuve D, et al. Patient perspectives on the role of community pharmacists for antidepressant treatment: A qualitative study. Canadian pharmacists journal : CPJ = Revue des pharmaciens du Canada : RPC. 2018;151(2):142-8. doi: 10.1177/1715163518755814
Reference 29: Humphries B, Collins S, Guillaumie L, Lemieux J, Dionne A, Provencher L, et al. Women's beliefs on early adherence to adjuvant endocrine therapy for breast cancer: A theory-based qualitative study to guide the development of community pharmacist interventions. Pharmacy (Basel, Switzerland). 2018;6(2):53. doi: 10.3390/pharmacy6020053
We have defined data saturation as follow: “This rigor criterion indicates that new interviews do not provide substantial additional information on the research question30-31.” (lines 138-140) and added information on how we evaluated that this sample size was sufficient to reach data saturation:. “Throughout the data collection process, we assessed saturation based on the interviewer’s (VT) summaries and preliminary data analysis, which were discussed with the principal investigator (SL). During this process, we found that 20 interviews were sufficient to reach data saturation” (lines 138-141).
Reference 30: Guest G, Bunce A, Johnson L. How Many Interviews Are Enough?:An Experiment with Data Saturation and Variability. Field Methods. 2006;18(1):59-82. doi: 10.1177/1525822x05279903
Reference 31: Miles MB, Huberman AM, Saldana J. Qualitative Data Analysis: A Methods Sourcebook. Fourth ed: SAGE Publications, Inc.; 2018.
In section "2.1 The SOIE program pilot RCT," while you have provided a concise summary of previous papers, further condensation might enhance clarity of the description.
We thank the Reviewer for this suggestion aiming to improve our manuscript. While we believe most of the information need to be retained in order for readers to understand the context of this qualitative study (e.g., study design, conceptual framework), we did our best to further condense to this section:
“2.1. The SOIE program pilot RCT
The SOIE program and its pilot RCT are described elsewhere20. Briefly, we conducted a pilot single-center, parallel-group RCT. In addition to usual care, the intervention group re-ceived SOIE, while the control group received usual care only. Women were recruited at their first AET prescription for non-metastatic breast cancer at the Breast Disease Center of the CHU de Québec-Université Laval. We included in the RCT women ≥18 years old, diagnosed with non-metastatic breast cancer, having a first AET prescription, fluent in French, and able to ac-cess the Internet and in-person program activities. Psychosocial factors hypothesized to influence AET adherence according to our conceptual model20 and targeted by the program (i.e. intention to persist with AET, attitude towards AET, subjective norm, perceived behavioral control, AET knowledge, perceived social support, coping planning, anticipated regret, and fear of recurrence) were measured through questionnaires validated by our team, before randomization, and after 3 and 12 months. Adherence was measured using questionnaires and pharmacy records. The study was approved by the Research Ethics Board (MP-20-2018-4131) and participants provided written consent.” (lines 84-98).
Reviewer 3 Report
Comments and Suggestions for Authors
very relevant study because it is important to ensure that complete treatment is being adopted by women who suffer from breast cancer.
I am not surprised with the results related to positive support between women but ? difficulty with those who are negative about it?
Recommendations for future studies is certainly an important direction
please ensure to review the comments inside the text
in particular in the rigor of the analysis and the selection process of those included in the qualitative part.

Author Response
AUTHORS’ RESPONSE (CURRONCOL-3390342)
Comments and Suggestions for Authors
—Reviewer 3
We thank the Reviewers for their helpful comments and suggestions. In the following, the Reviewers’ comments are reproduced integrally, are numbered, and appear in bold text. Our responses follow each comment. Any text cited directly from the manuscript is in quotation marks, while any changes made to the text submitted initially are in italics. Changes made to the manuscript in response to Reviewers’ comments and suggestions are shown in track changes in a revised version of the manuscript.
Very relevant study because it is important to ensure that complete treatment is being adopted by women who suffer from breast cancer.
- I am not surprised with the results related to positive support between women but ? difficulty with those who are negative about it?
As reported in our results, most women benefited from being in contact with others who had lived through a similar experience (see section “Support from healthcare providers and peers”). However, we believe it is important to acknowledge that for a minority, the group activities caused feelings of insecurity or concerns when there were participants who shared particularly negative experiences. In the discussion, we recommended developing a more detailed plan to manage the sharing of negative experiences during group activities (lines 315-316).
- Recommendations for future studies is certainly an important direction
Thank you for your comment. In the Conclusion, we indeed pointed out three different areas for future studies: 1) the development of multifaceted interventions to enhance AET adherence (lines 358-361); 2) the evaluation of long-term benefits of programs such as SOIE (lines 365-367), and 3) the development and validation of new scales to assess the psychological and emotional benefits of programs such as SOIE (lines 367-371).
- please ensure to review the comments inside the text in particular in the rigor of the analysis and the selection process of those included in the qualitative part.
3.a Can you briefly describe usual care?
In this section, we refer to the article presenting the SOIE program and the pilot RCT (reference: Assan O, Memoli V, Guillaumie L, Turcotte V, Lemay M, Dionne A, et al. Pilot randomized controlled trial of a program to enhance experience and adherence with adjuvant endocrine therapy among women with non-metastatic breast cancer: 12-month quantitative results. J Cancer Surviv. 2024:Epub ahead of print. doi: 10.1007/s11764-024-01599-y). Usual care is described in detail in this article: “Usual care consisted of information provided by the oncologist at the time of AET prescription and follow-up visits (usually, every 6 months in the first year) and occasionally providing written information and reference to a nurse navigator or other multidisciplinary team members when required. Patients typically refill their AET on a monthly basis at their community pharmacy and receive information and advice about the medication from their community pharmacists.” We prefer that readers have access to the full description of usual care rather than presenting only selected elements in the submitted manuscript, which would provide partial information.
3b. Clarify your recruitment as you did provide numbers below?
We have added more details on our recruitment approach for this qualitative study: “We selected potential participants for this qualitative study to ensure that the proportion of women with certain characteristics (e.g., age, type of AET) was similar to that of the entire sample in the pilot RCT. Additionally, we purposively sampled women with varying levels of satisfaction with the program, as reported in their questionnaires, to capture a diverse range of perspectives.” (lines 104-108). In addition, we have specified how many women were assigned to the intervention group in the pilot RCT and were eligible for this qualitative study: “Women were selected among the 48 women who completed their one-year follow-up based on a purposeful sampling22 that considered their responses to the questionnaires (lines 103-105).
3c. What about describing or adding clinical saturation?
We have added details about saturation. Please see the following response we provided to Comment #2 from Reviewer # 2:
We have added details on the initial estimation of our sample size: “Based on our previous experience in qualitative research on medication use among cancer patients26-29, we initially estimated that 20 interviews would be sufficient to achieve data saturation.” (lines 136-138).
We have defined data saturation as follow: “This rigor criterion indicates that new interviews do not provide substantial additional information on the research question30-31.” (lines 138-140) and added information on how we evaluated that this sample size was sufficient to reach data saturation:. “Throughout the data collection process, we assessed saturation based on the interviewer’s (VT) summaries and preliminary data analysis, which were discussed with the principal investigator (SL). During this process, we found that 20 interviews were sufficient to reach data saturation” (lines 138-141).
3d. how did you ensure rigour in your data analysis?
In the manuscript initially submitted, we described how the coding process was performed: “In collaboration with the principal investigator (SL— Anthropology and Epidemiology), the research professional carried out data segmentation and categorization from the first four interviews and elaborated a preliminary codebook using a mixed approach (inductive and deductive)24. The development of this preliminary codebook was based on the interview guide, the conceptual model of the SOIE program and allowed the emergence of codes from the corpus25. A validation exercise was performed in which a research assistant (GB or AL—Psychology) independently proceeded to the data categorization from excerpts of the same four interviews using the codebook. In case of disagreements, the team discussed the categorization until consensus. The codebook was then refined with the collaboration of senior researchers (SL and LGuillaumie—Patient education). This process was carried out eight additional times until all interviews were analyzed”. (lines 119-129)
Following this comment form the Reviewer, we added information on how the analysis and interpretation was performed: “The research professional (VT) and research assistants (GB, AL) created summaries for each code, including exemplary quotes, as well as comparative tables synthesizing the main codes for women and HCPs. These summaries and the interrelationships between codes were discussed during meetings with the research team (SL, L. Guillaumie) and guided the interpretation of the data.” (lines 130-133).
3e. This aspect is unclear! How did you determine this
These characteristics were determined from the pilot RCT questionnaires: “Of the 21 women and the 9 HCPs invited to participate in an individual interview, 20 and 7 agreed, respectively. Their characteristics, collected through the pilot RCT questionnaires, are presented in Tables 1 and 2” (lines 147-148).
3f. Should have been clarified before in the text… and in why did you choose these women in particular? Limit the information?
As mentioned in response to Reviewer’s comment # 3b, we have further specified how was performed the selection for the qualitative study: “We selected potential participants for this qualitative study to ensure that the proportion of women with certain characteristics (e.g., age, type of AET) was similar to that of the entire sample in the pilot RCT. Additionally, we purposively sampled women with varying levels of satisfaction with the program, as reported in their questionnaires, to capture a diverse range of perspectives.” (lines 104-110).
3g. You have great conclusion but can you add it as recommendation because your results are interesting!
Thank you for your positive feedback on our work. In our conclusion, we highlighted three areas for future research or clinical development: 1) the development of multifaceted interventions to enhance AET adherence; 2) the evaluation of the long-term benefits of programs such as SOIE; and 3) the development and validation of new scales to assess the psychological and emotional benefits of programs like SOIE. As these suggestions are based on a qualitative study derived from a pilot RCT, we prefer at this stage to present them as potential next steps rather than making strong recommendations
Round 2
Reviewer 1 Report
Comments and Suggestions for Authors
The authors have taken into account all the suggestions, so the article is ready to be published.
Reviewer 2 Report
Comments and Suggestions for Authors
Thank you for appropriately correcting the items I pointed out. I have no further comments on this paper.
Reviewer 3 Report
Comments and Suggestions for Authors
thanks for your comments and changes
as they certainly clarify my questions